# A More Comfortable Method for Hydrostatic Weighing: Head above Water at Total Lung Capacity

**DOI:** 10.3390/jfmk9010041

**Published:** 2024-02-28

**Authors:** Erin White, Silas Bergen, Annabelle Berggren, Lillian Brinkman, Brianna Carman, Lucas Crouse, Emma Hoffmann, Sara Twedt

**Affiliations:** 1Department of Health, Exercise and Rehabilitative Sciences, Winona State University, Winona, MN 55987, USAlillian.brinkman@go.winona.edu (L.B.); brianna.carman@go.winona.edu (B.C.); lucas.crouse@go.winona.edu (L.C.);; 2Department of Mathematics and Statistics, Winona State University, Winona, MN 55987, USA; sbergen@winona.edu

**Keywords:** fitness, body composition, comparison, underwater weighing, percent body fat

## Abstract

Hydrostatic weighing (HW) requires full submersion with the lungs at residual volume (RV) which is uncomfortable. Therefore, the purpose of this study was to find a more comfortable way to complete HW. A HW system was used to complete three comparisons: comparison 1: change in head position (head above water vs. head below water (HAW vs. HBW)), comparison 2: change in lung volume (total lung capacity (TLC) vs. RV), and comparison 3: change in head and lung volume changes. Participants were separated by males (*n* = 64) and females (*n* = 58). Comparison 1: HAW resulted in higher mean percent body fat (PBF) than HBW (4.5% overall, 3.8% in males, 5.4% in females, *p* < 0.05). Comparison 2: TLC resulted in lower mean PBF than RV (5.1% overall, 5.3% in males, 4.8% in females, *p* < 0.05). Comparison 3: HAW@TLC resulted in significantly lower (1.5% lower, *p* = 0.003) mean PBF for males but was not significantly lower for females or overall (0.6% higher, *p* = 0.39, 0.6% lower, *p* = 0.18, respectively) compared to HBW@RV. In conclusion, keeping the head above water and taking a deep inhale makes HW a more enjoyable, and accessible experience for everyone while still producing accurate PBF results.

## 1. Introduction

While body weight is tracked throughout the lifespan, body composition is the more important indicator of overall health. High percent body fatness is associated with negative health implications such as hypertension, diabetes, cancer, and cardiovascular disease. Low percent body fatness is associated with malnutrition, osteoporosis, osteopenia, and muscle wasting [1]. Knowing body composition can assist in tracking training goals and interpreting overall physique; however, every method of body composition analysis poses unique advantages and disadvantages. Field methods of determining body composition include anthropometric measurements such as body mass index (BMI), waist circumference, waist to hip ratio, skinfolds, single frequency bioelectrical impedance analysis (BIA), and multifrequency BIA [2,3,4]. Lab methods of determining body composition include hydrostatic weighing (HW), air displacement plethysmography (BODPOD), isotope dilution method, dual-energy Xray absorptiometry (DEXA), computed tomography (CT), computed tomography body composition, magnetic resonance imaging, and whole-body potassium counter [2,3,4]. With an array of options, deciding which method to use depends on accessibility of the equipment, one’s financial means, and the degree of accuracy desired. DEXA is the clinical gold standard and while comfortable, the procedure emits radiation that poses risk to the participant. It is also the most expensive piece of equipment. Due to this limitation, HW has been a trusted and valid method of determining body composition for decades within laboratory settings [5]. The accepted standard of performing HW requires full submersion of the participant while at residual volume (RV). This can cause physical and psychological discomfort which can deter participation as it is an unnatural and uncomfortable feeling [6]. Therefore, previous groups have studied ways to minimize these difficulties and discomforts while still maintaining the validity of HW.

RV is the standard procedure because it is the lung volume least affected by hydrostatic pressure. Total lung capacity (TLC) is most affected by hydrostatic pressure on the lungs; however, Weltman and Katch [7] found that HW at TLC required fewer trials compared to RV. Limiting time in the water by fewer total trials, along with using a more natural lung volume is beneficial in making participants feel more comfortable about the body composition assessment. It has also been shown by previous studies that full body submersion is not needed. Israel et al. [8] found that keeping the head above water is a valid alternative to full submersion in morbidly obese females. Donnelly et al. [9] also found that due to increased buoyancy, 25% of obese participants could not fully submerge. Developing more comfortable ways to accurately measure body composition via HW will allow access to a wider and more diverse population of participants. Recently, Tesch et al. [6] has developed a head volume prediction (HV_PRED_) equation with the use of anthropometric measurements, so head submersion is not necessary. However, this equation needs further validation. Due to the discomfort caused by the gold standard of hydrostatic weighing and the simplicity of incorporating head volume measurements, the purpose of this study was to find a more comfortable way to complete HW. To accomplish this, three tests were done: (1) the concordance of HV_PRED_ equations [6] with head submersion, (2) the validity of TLC using RV during HW, and (3) the validity of HAW@TLC (the most comfortable technique) using HBW@RV (the most unnatural and uncomfortable technique).

## 2. Materials and Methods

### 2.1. Participants

All participants were volunteers aged 18 years old or older, not pregnant, did not possess pacemakers, and were free of pulmonary diseases. This investigation was conducted at the Health, Exercise, and Rehabilitative Sciences Laboratory at Winona State University in Winona, MN using a convenient sample. Recruitment methods primarily included posters hung within the university and local community places and word-of-mouth. All participants were instructed to come to the laboratory for their 30-min session wearing minimal, tight-fitting clothing. There were no restrictions for eating or exercise prior to data collection. The study was conducted according to the guidelines of the Declaration of Helsinki and approved by the Institutional Review Board of Winona State University (1987424-4, approved 21 March 2023). Informed consent was obtained from all participants involved in the study prior to participation. Each participant was given a copy of the informed consent for their records. Participants then self-recorded their age, sex, preferred pronouns, and physical activity level rating, using the Saltin-Grimby Physical Activity Level Scale (SGPALS) [10,11,12].

### 2.2. Anthropometric Measurements

#### 2.2.1. Height & Weight

Each participant was asked to void their bladder if they had not voided within the last 30 min prior to their appointment. Height was measured using a Seca wall-mounted stadiometer. If the participant had long hair, the hair was tied back into a low ponytail. Height was recorded to the nearest 0.1 cm. Participants were instructed to remove any jewelry, watches, glasses, or other accessories before having their weight recorded. All dry weight measurements were taken in the minimal, tight-fitting clothing that the participant intended on wearing in the weighing tank. A calibrated Health O’Meter Professional balance beam scale was used to measure dry body weight in kilograms (kg). All weights were recorded to the nearest 0.1 kg.

#### 2.2.2. Head & Face Girth

Head measurements of each participant were taken by trained technicians which included head girth and face girth, as described by Tesch et al. [6]. To standardize the procedure, all participants were seated, put their hair into a low ponytail (tucked behind ears) if needed, held their head in the Frankfurt Horizontal Plane, and did not move their jaw while being measured. An AnthroFlex steel metric tape measure was used for both head measurements using the cross-hand technique recommended by the International Society for the Advancement of Kinanthropometry (ISAK). During the head measurements, participants viewed a short instructional video to become familiar with the in-water procedures.

Head girth (HG) measures the maximal horizontal circumference of the head spanning from the glabella to the furthest protruding point of the cranium [6]. The technician stood on the right side of the seated participant and a second technician moved around all sides of the participant to ensure that the tape was level at all points around the head (Figure 1). Two measurements were taken, measured to the nearest millimeter (mm) by two different trained technicians. A third measurement was taken if the measurements between the technicians differed by more than 5 mm. On the rare occurrence no two measurements were within 5 mm, a fourth trial was taken. The average of two HG measurements within 5 mm was used in predicted head volume equations (see Equations (1) and (2)).
Males: HV_PRED_ = 0.1294 ⋅ HG + 0.0299 ⋅ FG + 0.0055 ⋅ MA − 5.7506 (1)
Females: HV_PRED_ = 0.1314 ⋅ HG + 0.0504 ⋅ FG+ 0.0094 ⋅ MA − 7.3181(2)

Face girth (FG) is the maximum circumference from under the chin to the vertex of the head [6]. The tape was placed immediately superior to the hyoid bone where the neck and chin meet and wrapped up the sides of the face (using the tragion as a landmark) to the vertex of the head. To ensure a proper measurement, the tape needed to be equidistant in relation to the tragion on each side of the head (Figure 2). FG was taken by two technicians, one standing behind the seated participant, with the second moving around the participant to ensure that the measuring tape was level and crossed all relevant landmarks. Two measurements were taken and measured to the nearest millimeter (mm) by two different trained technicians. A third measurement was taken if the measurements between the technicians differed by more than 5 mm. On the rare occurrence no two measurements were within 5 mm, a fourth trial was taken. The average of two FG measurements within 5 mm was used in predicted head volume equations (see Equations (1) and (2)).

### 2.3. Hydrostatic Weighing

Each participant showered to wash off any lotions or deodorants prior to entering the water tank. The tank used was approximately 3.5 ft by 4.75 ft and 3.5 ft in depth and water temperature was maintained between 31–34 °C. The participant then entered the tank to be weighed. The computer program used was designed specifically for hydrostatic weighing (HW) (HydroDensity software version HD2, EXERTECH^®^—Dresbach, MN, USA). Instructions for the hydrostatic weighing procedure were verbally explained in detail to each participant. Participants were asked to brush off any air bubbles that were trapped in their tight-fitting clothing or hair. Participants first completed total lung capacity (TLC) trials with the head above water (HAW) and the head below water (HBW). Due to the increased buoyancy with TLC, all participants wore a 2.1 kg weighted belt worn around the waist. They were instructed to keep their shoulders below the level of the water to prevent overloading the weighing system and reduce perturbation of the water for the most accurate results. A nose clip was secured to each participant to prevent air leakage through the nasal passage. For TLC trials, the participants were instructed to position their head so the tips of the earlobes and the inferior surface of the chin were in line with the level of the water (Figure 3). The participants inhaled as much as they possibly could and were instructed to hold this position above water for 3 s. Following the 3 s above water, the participants bent forward at the waist to fully submerge themselves below the water at the same lung capacity. Once below the water, the participants remained underwater for 5 s on their count. This procedure constituted one trial, where their weight was measured with their head above and below the water at the exact same lung volume (TLC). Participants completed hydrostatic weighing at TLC until 3 trials within 100 g (for HAW and HBW) were recorded with a maximum of 7 trials. Technicians recorded the participant’s weight for HAW and HBW in kilograms from the EXERTECH^®^ computer system by selecting 100 ± 2 samples which best represented the mean weight with head above and head below water [6]. The average of three weights, for both HAW and HBW, within 100 g were used for analysis.

To minimize the effect of fatigue on the participant, residual volume (RV) trials were taken after TLC trials. The weight belt was removed, but the nose clip remained. One RV trial consisted of the participant being instructed to fully exhale, bend forward until the head was totally submerged, continue to exhale as much air as possible out of the lungs, and to remain underwater until they physically could not. The technicians recorded underwater weight from the EXERTECH^®^ computer system by selecting 100 ± 2 samples which best represented the mean weight underwater at RV [6]. A minimum of 3 and maximum of 5 trials were performed for head below water at RV (HBW@RV). The number of trials was capped at 5 to minimize fatigue, yet still reap the benefits of a learning effect from performing the previous trials. The average of three weights within 100 g was used for analysis. Once RV trials were finished, the participant was completed with the study and had the option to learn their percent body fat (PBF)% from the trials at RV if they chose to.

Head volumes were predicted from the equations of Tesch et al. [6] (Equations (1) and (2)) and lung volumes were calculated from the equations of Quanger et al. [13] for TLC (Equations (3) and (4)) and RV (Equations (5) and (6)). Body density (Equations (7)–(10)) was calculated based on the principle of Archimedes [3] with adjustments for head position and lung volume. The newly published equation by Tesch et al. [6] (Equation (7)) was used which accounts for the predicted head volume (HV_PRED_); however, because the lung volume also changed, RV was replaced with TLC (Equation (8)) to calculate body density for the HAW condition. For the submersion trials, given that the lung volume changed by condition, either TLC or RV were replaced in the equation where appropriate (Equations (9) and (10)). PBF was calculated by use of the equation of Brozek et al. [14] (Equation (11)) with the corresponding body density from each condition.
Male: TLC (L) = 7.99 ⋅ Height (m) − 7.08 (3)
Female: TLC (L) = 6.60 ⋅ Height (m) − 5.79 (4)
Male: RV (L) = 1.31 · Height (m) + 0.022 ⋅ Age (years) − 1.232 (5)
Female: RV (L) = 1.812 ⋅ Height (m) + 0.016 ⋅ Age (years) − 2.003 (6)
Db_HAW(HV)@RV_ = MA ⋅ ((MA − MW_HAW_) ⋅ DW^−1^ + HV_PRED_ − RV − 0.1)^−1^(7)
where, MA = Dry Body Mass in Air, MW_HAW_ = Mass in Water with the Head Above Water; DW = Density of Water
Db_HAW(HV)@TLC_ = MA ⋅ ((MA − MW_HAW_) ⋅ DW^−1^ + HV_PRED_ − TLC − 0.1)^−1^
(8)
Db_HBW@TLC_ = MA ⋅ ((MA − MW_HAW_) ⋅ DW^−1^ − TLC − 0.1)^−1^
(9)
Db_HBW@RV_ = MA ⋅ ((MA − MW_HAW_) ⋅ DW^−1^ − RV − 0.1)^−1^
(10)
PBF = (4.570 ⋅ Db^−1^ − 4.142) ⋅ 100 (11)

### 2.4. Statistical Analysis

The statistical software SPSS Version 29.0.1.0 (171) (IBM) was used to perform all analyses. All analyses were completed with the total data set and separated by self-reported sex. Means, standard deviations (SD), and frequencies were used to describe the participants’ physical characteristics. Bland-Altman plots were created for each comparison because the popular use of correlation coefficients has been found to be unreliable [15,16]. The three different comparisons were: comparison 1: comparing estimated PBF due to change in head position (head above water at total lung capacity (HAW@TLC) versus head below water at total lung capacity (HBW@TLC)), comparison 2: comparing estimated PBF due to change in lung volume (HBW@TLC and head below water at residual volume (HBW@RV)), and comparison 3: comparing estimated PBF due to change in head position and lung volume (HBW@RV and HAW@TLC). Means and SDs of each comparison were computed, as well as Bland-Altman plots to investigate whether any differences were proportional to PBF. A linear regression was used to determine if any disagreement between the two measurements was proportional to PBF. The level of significance for the linear regressions for proportional bias was set at α = 0.05.

A paired samples *t*-test was used to test whether the mean difference of each comparison (change in head position only, change in lung volume only, or change in both) differed significantly from 0. Paired samples *t*-tests were also used to compare the average of the first and second consistent head and face girth measurements within 5 mm. The level of significance was set at α = 0.05.

Lin’s Concordance Correlation Coefficient (LCCC), which evaluates the agreement between two measurements, was computed to evaluate each comparison [17,18]. Lin’s CCC ranges from 0 to ±1 and can be interpreted similar to a Pearson r correlation in which 1.00 is perfect concordance, 0.80–0.90 is a very strong concordance, 0.60–0.70 is moderate concordance [19].

## 3. Results

Physical characteristics, separated by sex, for 122 participants are shown in Table 1. The complete data set was used for analysis except for two individuals in which the participants did not follow the RV methods, disqualifying their data, therefore, all RV data *n* = 120. Sex, preferred pronouns, age, ethnicity, and SGPALS score were self-reported. The majority of the sample was White (88.5%) between the ages of 18 and 24 (84.4%) who were physically active, defined as a SGPALS score of 3 or 4 (76.2%). Height, weight, head girth, and face girth were measured. Out of the 122 participants, only 10 (8.2%) required a third head girth measurement and none required a fourth. Regarding face girth, 24 participants (19.7%) required a third face girth measurement and only one participant required a fourth. Table 2 depicts the means and standard deviations of the first and second consistent HG and FG measurements. The measured HG and FG were not significantly different between the two trials within 5 mm (Table 2). When selecting the samples that best represented the participant’s underwater weight the mean ± SD number of samples selected for HAW@TLC was 100.05 ± 1.13, HBW@TLC was 100.24 ± 1.18, and HBW@RV was 100.26 ± 1.08. While the minimum number of samples selected throughout the study was 96 and the maximum was 109, over 95% of the samples for each trial were within the range of the goal of 98–102 samples (100 ± 2).

In comparison 1, the only variable changed was head position (HBW@TLC vs. HAW@TLC), which specifically tests the accuracy of the PBF w/HV_PRED_ equation suggested by Tesch et al. [6]. Table 3 shows that HAW@TLC resulted in statistically significantly higher mean PBF measurements than HBW@TLC, both overall and separately for males and females. The Bland-Altman plots (Figure 4A–C) illustrated no evidence that the difference in calculated PBF between head position was proportional, both overall and separately for males and females, as there was no significant relationship between the difference in these metrics and the average of the metrics (*p*-value for proportional relationship > 0.05). The LCCC values were >0.8, indicating the percent body fat values have very strong concordance, both overall and separately for males and females, when using the standards of interpretation for Pearson’s correlation coefficient (Figure 5A–C).

In comparison 2, separately for males and females (Table 3). The Bland-Altman plots (Figure 4D–F) demonstrated that the difference in calculated PBF with a change in lung volume was statistically proportional for females (*p*-value for proportional relationship = 0.019, Figure 4F), but not for males (*p*-value for proportional relationship = 0.236, Figure 4E), thus driving the overall proportional bias (*p*-value for proportional relationship = 0.012). Specifically, overall and for females, HBW@RV appeared to yield higher PBF estimates than HBW@TLC for people with less body fat, and lower estimates for people with higher body fat. Figure 5D–F further illustrate that these two measures had the greatest amount of discordance with Lin’s CCC between 0.72–0.77, demonstrating that changing lung volume affects calculated percent body fat, even with the head below water.

In comparison 3, both head position and lung volume changed (HBW@RV and HAW@TLC). HAW@TLC yielded statistically significantly lower mean PBF measurements for males (*p*-value = 0.003) but no statistically significant difference overall (*p*-value = 0.175) or for females (*p*-value = 0.389, Table 3). Although there was a statistically significant difference for males, this difference was only 1.5% lower mean PBF from HAW@TLC; this difference is not clinically significant. The Bland-Altman Plot for males (Figure 4H) demonstrated there was no evidence that the difference in calculated PBF with a change in lung volume and head position was statistically significantly related to PBF (*p*-value for proportional relationship = 0.347). Figure 4I seemed to suggest proportional disagreement in females’ PBF from HBW@RV and HAW@TLC (*p*-value for proportional relationship = 0.017); however, this statistically significant relationship was driven by an outlying participant with low PBF. Omitting this outlier resulted in a no-longer-significant relationship between the difference in these estimates and the average PBF (*p*-value with outlier removed = 0.055). In Figure 5G–I LCCC values are >0.8 overall and separately for males and females, indicating the percent body fat values have very strong concordance when using the standards of interpretation for Pearson’s correlation coefficient.

## 4. Discussion

Comparison 1 specifically tested the accuracy of the HV_PRED_ equations developed by Tesch et al. [6] by looking at the difference in PBF with HAW and HBW while lung volume remained constant. The results of comparison 1 show that head position does matter when measuring PBF at TLC. As the results stated, HAW@TLC produced higher PBF values when compared to HBW@TLC, both overall and separately for males and females. The statistically significant (*p* < 0.05) differences in mean PBF in this study are contradictory to past studies that compared HBW and HAW weighing.

A study done by Evans et al. [20] revealed that mean PBF with the HAW was higher than mean PBF with the HBW by a mean difference of 0.66% (*p* > 0.05). Similarly, Israel et al. [8] found that PBF with the HAW was 0.66% higher than PBF with the HBW (*p* > 0.05). A study done by Donnelly et al. [9] found a 0% mean difference in PBF in males and a 0.7% (HAW higher) mean difference in females. As shown in the studies by Evans et al. [20], Israel et al. [8], and Donnelly et al. [9] no statistically significant difference in PBF was found when changing head position and keeping lung volume the same. However, a study completed by Heath et al. [21] found that PBF with the HAW was lower than PBF with the HBW by 2.8% in females (*p* < 0.0001) but only 0.1% lower in males. The mean difference in PBF was found to be statistically significant in females but not in males. Demura et al. [22] found a mean difference in PBF of approximately 5% higher in HAW. These results were statistically significant. Lastly, the study done by Tesch et al. [6] found no statistically significant differences in the male experimental group, male validation group, female experimental group, or female validation group.

From the previously listed studies, only two out of the six populations produced statistically significant differences in PBF. The low number of statistically significant results contradict the present study’s finding of statistically significant results. However, five out of the six listed populations were niche populations. For example, three out of the six populations included only individuals who were female and morbidly obese. The majority of these studies also used differing equations, indicating that an accepted equation for the general population is yet to be determined. In 1988, Donnelly et al. [9] developed equations for HAW weighing that produced non-statistically significant results. However, when Demura et al. [22] sought to validate these equations a statistically significant difference in PBF of approximately 5% was found. As it has been mentioned, the present study used equations developed by Tesch’s group [6]. The lack of statistically significant results from Tesch’s study [6] contradict the present study. However, these equations were both produced and validated within the same study. The validation group for Tesch’s study [6] was half the size of the population of the present study. Tesch et al.’s [6] validation group consisted of 21 males and 24 females, whereas the present studies population consisted of 64 males and 58 females. The difference of statistically significant results between two studies using the same equation is similar to the occurrence of Demura et al. [22] validating Donnelly et al.’s [9] equations and finding a 5% difference.

Comparison 2 specifically tested the measurement of PBF at different lung volumes (TLC vs. RV) with the head fully submerged. In the current study, it was found that TLC resulted in lower PBF by approximately 5%, and that this disagreement was roughly equivalent for both males and females. These findings contradict previous research such as Weltman and Katch [7] who observed a mean difference in percent body fat between TLC and RV of 0.9% for females and 0.5% for males (*p* = 0.05), both of which are within the measurement error for HW of ±1.5% [14]. Another study by Warner et al. [23] found that TLC compares favorably to RV, with statistically significant differences (*p* = 0.001), as the differences in the methods were not clinically significant (PBF = RV: 16.34%, TLC: 15.47%). Latin and Ruhling [24] observed a mean difference in PBF of 1.1% between RV and TLC. This is statistically significant (*p* < 0.05) but the difference is not clinically significant. Overall, the current study contradicts previous research because it was found that lung capacity plays a significant role when measuring PBF.

Comparison 3 evaluated the most comfortable method of HW (HAW@TLC) to the gold standard which is the least comfortable method of HW (HBW@RV). In this comparison, both head position and lung volume changed. According to the current results, comparison 1 resulted in a higher PBF from HAW for all participants by approximately 5%. Comparison 2 then resulted in a lower PBF from TLC for all participants by approximately 5%. When HAW was paired with TLC in comparison 3, there was no significant difference compared to the gold standard. Therefore, the current results suggest that when using the HV_PRED_ equation [6] with TLC, the equation produced an accurate measurement of PBF compared to the gold standard (HBW@ RV).

The findings from comparison 3 indicate that this more comfortable method of HW is an acceptable method for determining body composition. This method increases participant comfort in the water by allowing the head to remain above water and inhale rather than exhale. Furthermore, it is simple and inexpensive to take head measurements, easy to cue participants into position without needing to adjust the scale up or down, and clear communication can be had with participants throughout the trials. Allowing full communication between the participant and technician also alleviates many of the anxieties participants often feel. The benefits of HAW@TLC allow a wider variety of individuals to participate in HW, including individuals who are obese and are not physically able to fully submerge [8,9].

Limitations of this study include the lack of diversity of participants, setting a maximum number of trials for each condition, and not using a spirometer to measure lung volumes. Despite efforts to recruit a diverse sample of participants, this study included participants that were mostly White, 18–24 years-old, and were physically active. Therefore, the generalizability of the current results is limited. Future research should include wider diversity in race, age, physical activity, and those with extremely low and high PBF. The maximum number of trials was set because the main goal was participant comfort. The TLC trials were capped at seven and RV trials were capped at five based off the study done by Bonge and Donnelly [25] which found that as many as 10 trials are not needed and the first three trials within 100 g provide accuracy. By capping the number of trials, it minimized participant fatigue while maximizing the learning effect to increase consistency in results. However, setting a maximum number of trials resulted in some participants not obtaining three consistent values within 100 g for each condition. In comparison 1, 27.0% (*n* = 33) of the participants achieved only two consistent measurements instead of three. In comparison 2, 24.6% (*n* = 30) achieved only two consistent measurements instead of three. In comparison 3, 1.7% (*n* = 2) did not achieve two consistent measurements, and 28.0% (*n* = 33) did not achieve a third consistent measurement. Therefore, it is possible the data could be more consistent; however, after extensive pilot testing, we did not notice an improvement in consistent results (participant got fatigued) and thus chose to cap at seven trials for TLC and five for RV. Another difference between previous research and this study was the use of a spirometer. Without the use of a spirometer in this study, there was a greater chance of human error in trusting that the participants followed protocol of maximally inhaling for TLC and maximally exhaling for RV. The use of a spirometer would further solidify the accuracy of the data collected in future studies, by measuring exact lung volumes used. But spirometers are not extensively used in lab-based settings; thus, we wanted to mimic what would happen in real life body composition testing. Adding the use of a spirometer would have been helpful; however, it would not mimic “typical” HW testing. Further, the acceptable criteria were measurements within only 100 g, therefore demonstrating the consistency of lung volumes.

While there were limitations, pilot testing for this study was extensive. Every procedure implemented was rigorously pilot tested to ensure the most accurate data possible while keeping participant comfortability in mind. All researchers were instructed by an International Society of the Advancement of Kinanthropometry (ISAK) member on how to properly take HG and FG measurements. With extensive practice, a technician can become proficient in measuring HG and FG using simple and affordable equipment. Tesch et al. [6] used five individual head measurements, however, only head girth and face girth showed the highest individual correlations when correlated with the mass of water displaced by the head. Therefore, these were the only two measurements the technicians learned and added only 2–3 min to the total data collection time for participants, making it a viable alternative to head submersion. Additionally, several lung volumes were pilot tested based off previous research, including functional residual volume (FRV) [5], TLC [26], and the gold standard RV. FRV was too difficult to standardize without the use of a spirometer. Several attempts during pilot testing were contrary to results by Thomas and Ethridge [5] which found no difference in PBF when using FRV and RV, and thus was not used in the study. However, based on the study of Timson and Coffman [26] TLC was demonstrated to be a viable alternative to RV and was chosen in the current study for the ability to standardize the lung volume with “a maximal inhale” verbal queue. The pilot data demonstrated more consistent PBF results when cueing for TLC than FRV and two participants were able to achieve consistent results in only three trials, 11 achieved it within four trials, 11 achieved it within five trials, and 31 achieved it in six trials, with 67 individuals needing all seven trials. When using RV, 25 participants achieved three consistent measurements within the first three trials, 34 participants required a fourth trial, and 59 required a fifth trial. From the pilot data, TLC and RV were most consistent which made sense since they are at the extreme ends of lung volumes and not somewhere in the middle. Additionally, asking participants to inhale and hold their breath before going under water is a much more comfortable and natural feeling; thus, TLC was used instead of FRV.

## 5. Conclusions

In conclusion, hydrostatic weighing is a well-known method of assessing body composition, but the gold standard method can cause physical and psychological discomfort which deters participation. Therefore, the purpose of the present study was to find a more comfortable method of HW. Using this more comfortable method with head above water and fully inflated lungs produced similar PBF results to the gold standard method of HW with full body submersion at RV. This is an important finding and has several practical implications. One example is using the HV_PRED_ equation with TLC in a university Exercise Physiology class where the observation of the experience/method may be more important pedagogically than the actual result of the test. The use of HV_PRED_ equation at TLC would allow the students to gain experience with HW and performing PBF calculations without a student having to go through the unnatural and uncomfortable process of emptying their lungs to RV. The second practical implication would be clinical settings. The gold standard of HW is uncomfortable and unnatural overall, but it is especially difficult for children, individuals who are obese, individuals with pulmonary diseases, and individuals who do not like submerging their heads. Keeping the head above water and taking a deep inhale makes HW a more enjoyable, and accessible experience for everyone while still producing accurate PBF results.

## Figures and Tables

**Figure 1 jfmk-09-00041-f001:**
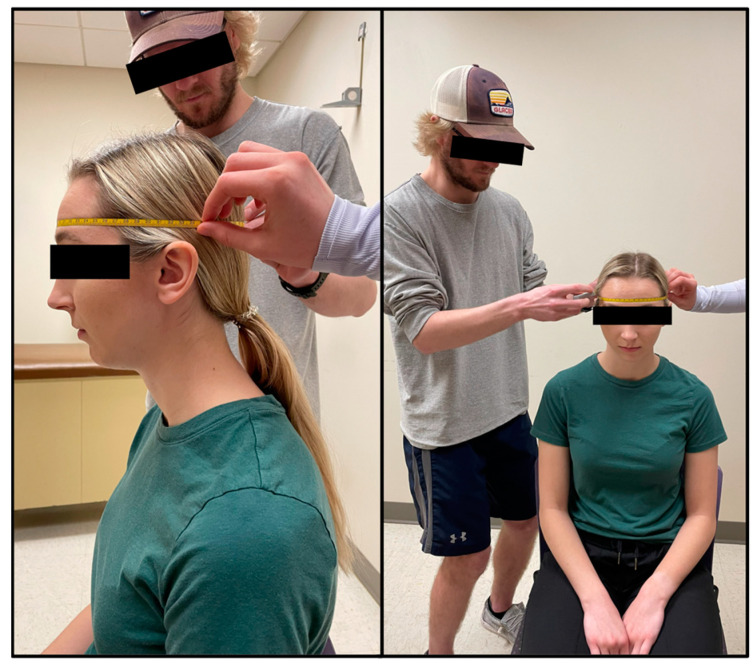
Head girth (HG) is the maximal horizontal circumference of the head spanning from the glabella to the furthest protruding point of the cranium.

**Figure 2 jfmk-09-00041-f002:**
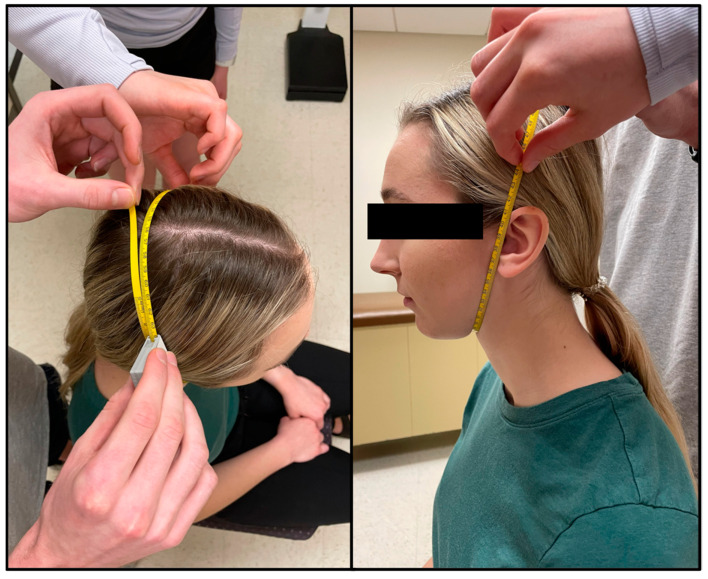
Face girth (FG) is the maximal circumference of the head spanning from the point immediately superior to the hyoid bone, where the chin meets the neck, to the vertex of the head while using the tragion as a landmark.

**Figure 3 jfmk-09-00041-f003:**
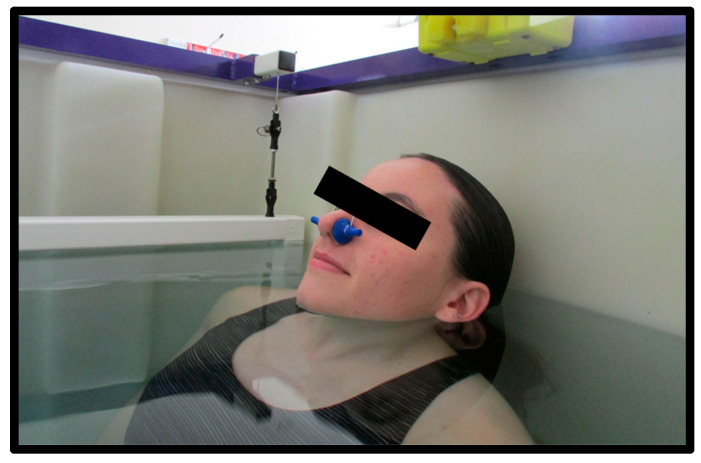
Head position for head above water (HAW) at total lung capacity (TLC). The inferior surface of the tip of the chin and the bottom of the earlobes are in contact with the surface of the water.

**Figure 4 jfmk-09-00041-f004:**
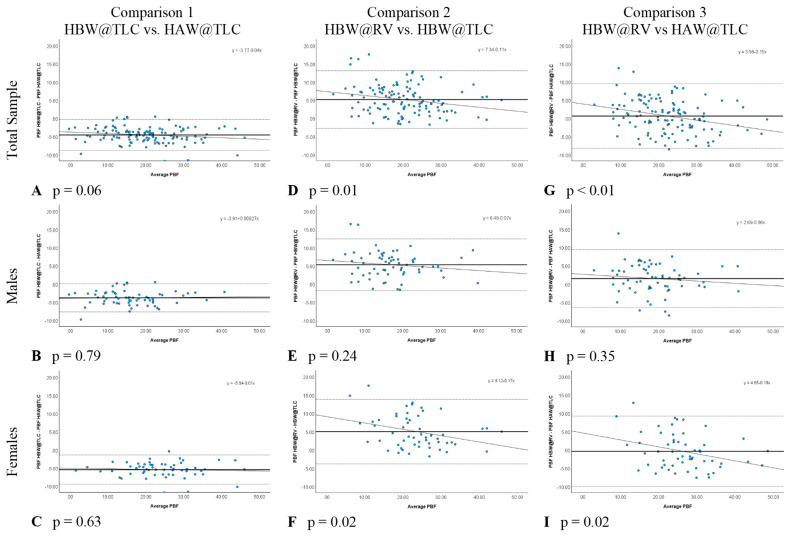
Bland-Altman Plots for each comparison with *p* values testing for proportional relationship. Each blue dot represents one participant. The solid line represents the mean difference in percent body fat (PBF) between the two conditions and the dotted horizontal lines represent the upper and lower 95% limits of agreement (±1.96 SD).

**Figure 5 jfmk-09-00041-f005:**
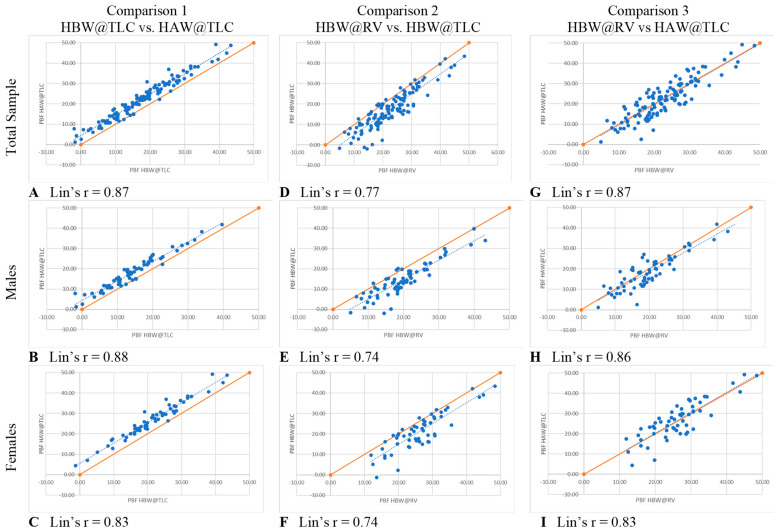
Lin’s Concordance Correlation Coefficients. Each blue dot represents one participant. Solid lines represent the line of perfect agreement; dashed lines represent lines of best fit.

**Table 1 jfmk-09-00041-t001:** Physical characteristics of the participants. To determine appropriate analysis groupings, the self-reported sex was used. Most participants identified with the corresponding pronouns; however, a couple of participants did not respond.

Characteristic	Male (*n* = 64)	Female (*n* = 58)	Total (*n* = 122)
Sex (%)			
Male	100.0	0	52.5
Female	0	100.0	47.5
Non-binary	0	0	0
Preferred Pronouns (%)			
He/Him	96.9	0	50.8
She/Her	0	100	47.5
They/Them	0	0	0
Age (yr) (mean ± SD)	25.3 ± 10.7	22.9 ± 8.4	24.1 ± 9.7
Age (yr) (min–max)	18–70	18–65	18–70
Ethnicity (%)			
White	87.5	89.7	88.5
Black/AA	3.1	1.7	2.5
Hispanic	1.6	3.4	2.5
Other	6.3	3.4	4.9
Height (cm) (mean ± SD)	182.1 ± 7.9	168.7 ± 6.2	175.7 ± 9.7
Weight (kg) (mean ± SD)	90.6 ± 14.7	68.5 ± 12.1	80.1 ± 17.4
BMI (kg/m^2^) (mean ± SD)	27.3 ± 4.1	24.1 ± 4.3	25.8 ± 4.5
SGPALS (%)			
1	1.6	3.4	2.5
2	4.7	36.2	19.7
3	32.8	41.4	36.9
4	59.4	17.2	39.3
Head Girth (cm)	58.3 ± 1.6	56.2 ± 1.6	57.3 ± 1.9
Face Girth (cm)	67.7 ± 2.1	63.0 ± 2.1	65.5 ± 3.1

Abbreviations: yr = year, SD = standard deviation, AA = African American, cm = centimeter, kg = kilogram, SGPALS = Saltin-Grimby Physical Activity Level Scale [10].

**Table 2 jfmk-09-00041-t002:** Participant head measurements and significance of consistency using paired *t*-tests.

	First Consistent Trial (mean ± SD)	Second Consistent Trial (mean ± SD)	Two-Sided *p*-Value
Males (*n* = 64)			
HG (cm)	58.3 ± 1.6	58.3 ± 1.6	0.91
FG (cm)	67.6 ± 2.1	67.7 ± 2.1	0.51
Females (*n* = 58)			
HG (cm)	56.2 ± 1.6	56.2 ± 1.6	0.40
FG (cm)	63.0 ± 2.1	63.0 ± 2.1	0.31

Abbreviations: HG = head girth, FG = face girth, SD = standard deviation, cm = centimeter.

**Table 3 jfmk-09-00041-t003:** Paired *t*-test results of the mean difference (*p* value) between each comparison.

	Comparison 1 HBW@TLC − HAW@TLC	Comparison 2 HBW@RV − HBW@TLC	Comparison 3 HBW@RV − HAW@TLC
Combined	−4.5 (<0.01)	5.2 (<0.01)	0.6 (0.13)
Males only	−3.8 (<0.01)	5.3 (<0.01)	1.5 (<0.01)
Females only	−5.4 (<0.01)	5.0 (<0.01)	−0.4 (0.56)

## Data Availability

Data requests will be reviewed by the corresponding author.

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
