# Peer review of "A More Comfortable Method for Hydrostatic Weighing: Head above Water at Total Lung Capacity"

_jfmk, 2024, doi:10.3390/jfmk9010041_

Round 1

Reviewer 1 Report

Comments and Suggestions for Authors

The publication presented for evaluation has a large application aspect, as it offers a convenient, simple and cheap method of hydrostatic weighing in water in order to calculate body composition, especially the level of fat, which is of great importance in monitoring health and sports.

In the introductory part, the authors present the methods used to determine body composition, highlighting their advantages and disadvantages, as a result of which they conducted research on various variants of determining body immersion with a full inhalation and holding the head, in and above the water.

They clearly propose an innovative, convenient method, presenting it as the goal of their work and proving its objective application

The advantage of the work is the rich research material and research methods used, consisting of anthropometric measurements, including height and body weight. Head and face circumference measurements are described very precisely. Hydrostatic weighing methods in the proposed research versions are described in equally detail.

The further part of the work regarding the statistical analyses, results, discussions and conclusions is unreserved.

In general, Paca, due to the proposed method of assessing body composition, is worth disseminating both in the theoretical and practical spheres.

Author Response

Thank you for your review.  The group of authors put in a tremendous amount of work and are very proud with the submitted manuscript.  We are glad to hear it was well-received.

Reviewer 2 Report

Comments and Suggestions for Authors

The manuscript 'A More Comfortable Method for Hydrostatic Weighing: Head Above Water at Total Lung Capacityis' is interesting, although several elements in this manuscript need to be improved before publishing:

The abstract should be corrected for better understanding. Using too many abbreviations makes it unreadable. Maybe it would be worth adding the last sentence from the conclusions.

The introduction should be strengthened with reference to more recent data methods of assessing body composition and with reference to the literature, e.g. in the lines: 33-36.

Materials and methods: Important details about the participants are missing, the age above 18 is too general - I suggest specifying the age range of the study participants, how they were recruited, and what their BMI was.

Results and Discussion: In addition, more factors should be taken into account when analyzing and discussing results, e.g. age, race, BMI.

The results, in my opinion, should take into account differences in ethnic groups or be analyzed only for the largest group. Does ethnicity influence the results? This should also be discussed in the discussion.

References: more new items need to be added

Author Response

The manuscript 'A More Comfortable Method for Hydrostatic Weighing: Head Above Water at Total Lung Capacityis' is interesting, although several elements in this manuscript need to be improved before publishing:

Thank you for your review and attention to detail, we have made revisions as deemed appropriate.  The responses are listed below and highlighted within the revised manuscript.

The abstract should be corrected for better understanding. Using too many abbreviations makes it unreadable. Maybe it would be worth adding the last sentence from the conclusions.

We agree, this was confusing and we were able to better word this.  It is much clearer now with what each comparison was.  As suggested, we also used the last sentence from the conclusion and used it in our concluding paragraph within the abstract.

The introduction should be strengthened with reference to more recent data methods of assessing body composition and with reference to the literature, e.g. in the lines: 33-36.

We appreciate your input and have strengthened this area to show both the field and laboratory methods of body composition (Lines 32-39).

Materials and methods: Important details about the participants are missing, the age above 18 is too general - I suggest specifying the age range of the study participants, how they were recruited, and what their BMI was.

This was on oversight on our part…this has been revised to further provide clarification for our recruitment procedures (lines 72, 75-76).  We did not address the age range at this point as there was only a minimum age to participate and no maximum age.  We did include the age range in the results.

Regarding age range and BMI, we added lines in Table 1 to further describe our participants as suggested.

Results and Discussion: In addition, more factors should be taken into account when analyzing and discussing results, e.g. age, race, BMI.

This is an excellent point.  Regarding race, our sample was highly homogeneous (88.5% White);

thus, when we looked at the three comparisons separately for whites and non-whites, and our results (significance of Comparisons 1 and 2, non-significance of Comparison 3) held for both subgroups. 

Regarding age, our sample was also highly homogenous (84.4% were between the ages of 18-24 yr) and when analyzing this subgroup alone, the results did not change.   

In addition, for your own curiosity, we did analyze it separately by activity status (SGPALS score 3 or more vs. score of <3) and it did not statistically significantly change the results.

We also reviewed the spread of body fatness and removed participants with less than the standard essential fat (3% for males and 12% for females) from the underwater trial at residual volume (the gold standard method).  This resulted in eliminating only 1 female participant, and again, did not alter the results.

We allude to these very issues in lines 397-399 “Future research should include wider diversity in race, age, physical activity, and those with extremely low and high PBF.”  to address these gaps within the literature.

The results, in my opinion, should take into account differences in ethnic groups or be analyzed only for the largest group. Does ethnicity influence the results? This should also be discussed in the discussion.

We agree, however, our sample was too homogeneous in ethnicity.  We do discuss it in lines 393-399 as part of our limitations as we also think body composition is affected by ethnicity, however, we did not have the power to analyze this.

In addition, as we were thoroughly reviewing our results, we noticed four typos in Table 3 and corrected them.  They do not alter the results any, but are now corrected.

References: more new items need to be added

We have added additional references to strengthen our introduction.
